# Microbial Communities in Retail Draft Beers and the Biofilms They Produce

Nikhil Bose,[a] Daniel P. Auvil,[a] Erica L. Moore,[a] Sean D. Moore[a]

[a]Burnett School of Biomedical Sciences, College of Medicine, University of Central Florida, Orlando, Florida, USA

**ABSTRACT** In the beer brewing industry, microbial spoilage presents a consistent threat that must be monitored and controlled to ensure the palatability of a finished product. Many of the predominant beer spoilage microbes have been identified and characterized, but the mechanisms of contamination and persistence remain an open area of study. Postproduction, many beers are distributed as kegs that are attached to draft delivery systems in retail settings where ample opportunities for microbial spoilage are present. As such, restaurants and bars can experience substantial costs and downtime for cleaning when beer draft lines become heavily contaminated. Spoilage monitoring on the retail side of the beer industry is often overlooked, yet this arena may represent one of the largest threats to the profitability of a beer if its flavor profile becomes substantially distorted by contaminating microbes. In this study, we sampled and cultured microbial communities found in beers dispensed from a retail draft system to identify the contaminating bacteria and yeasts. We also evaluated their capability to establish new biofilms in a controlled setting. Among four tested beer types, we identified over a hundred different contaminant bacteria and nearly 20 wild yeasts. The culturing experiments demonstrated that most of these microbes were viable and capable of joining new biofilm communities. These data provide an important reference for monitoring specific beer spoilage microbes in draft systems and we provide suggestions for cleaning protocol improvements.

**IMPORTANCE** Beer production, packaging, and service are each vulnerable to contamination by microbes that metabolize beer chemicals and impart undesirable flavors, which can result in the disposal of entire batches. Therefore, great effort is taken by brewmasters to reduce and monitor contamination during production and packaging. A commonly overlooked quality control stage of a beer supply chain is at the retail service end, where beer kegs supply draft lines in bars and restaurants under nonsterile conditions. We found that retail draft line contamination is rampant and that routine line cleaning methods are insufficient to efficiently suppress beer spoilage. Thus, many customers unknowingly consume spoiled versions of the beers they consume. This study identified the bacteria and yeast that were resident in retail draft beer samples and also investigated their abilities to colonize tubing material as members of biofilm communities.

**KEYWORDS** beer, bacteria, yeast, biofilm, *Acetobacter*, *Lactobacillus*, *Fructilactobacillus*

**Ad Hoc Peer Reviewer** Caleb Levar

Address correspondence to Sean D. Moore, sean.moore@ucf.edu.

Beer production involves controlled fermentation of plant sugar extracts in the presence of flavoring compounds to generate desirable beverages. During the brewing process, substantial effort is given to minimize exposure to contaminant microbes that compete for resources and impart undesirable flavors. In addition to careful fermentation, many finished beers are also filtered or pasteurized to further improve product stability, sometimes at the cost of product flavor quality. Unfortunately, these efforts are less effective if a beer is contaminated and spoiled during the packaging, distribution, or dispensing stages. On the dispensing end, we observed that routine draft line cleaning procedures are insufficient to maintain beer quality in retail draft systems and

that resilient microbial biofilms persist that rapidly reestablish complex spoilage communities. To begin to address this issue, we characterized microbial communities obtained from commercial draft beers and monitored their populations after they established biofilms during lab culturing.

In the nineteenth century, Louis Pasteur established that certain yeasts could be isolated and used to produce wines and beers with consistent and desirable characteristics (1). With those studies also came the discovery that beer and wine spoilage was caused by different microbes that competed for food resources and generated undesirable metabolites, such as lactic and acetic acid (1). Thus, the industry of fermented beverage production rapidly shifted away from so called "wild" inoculations and industry standards were put in place to carefully control and monitor the presence of both desirable and undesirable microbes (2). In the last few decades, the craft beer industry has revisited the use of alternative microbes and combinatorial culturing to greatly expand the style range and flavor profiles (3–6). With some irony, one goal of these efforts is to create products with scent and flavor complexities that match wild-fermented ales and lambics (7, 8). Nevertheless, great care and expense is still applied to minimize contamination by spoilage microbes and to ensure product stability (2, 9, 10).

Spoilage microbes enter the brewing process primarily from the addition of non-sterile ingredients, air exposure, or contaminated equipment (5, 11, 12). Several spoilage microbes are well known to the brewing community because they are commonly encountered and present a consistent threat; among these lactic acid bacteria (LAB), acetic acid bacteria (AAB), and wild yeasts represent dominant cohorts (10, 12–15). Interestingly, these types of microbes are also present as desirable members of the microbial communities found in wild fermentations, wherein they can improve flavor balance and impart sour characteristics as the beers are aged to maturity (16–18). In these aging processes, groups of microbes overtake one another to dominate the community in a cascading fashion, with each group consuming old metabolites and creating new ones. In addition, members of a microbial community can exhibit synergistic or antagonistic relationships with each other, which promotes unpredictable community restructuring depending on the metabolic and combat capabilities of the founding members (19–23). The transitions through community structures are a key feature that provides unique complexity to the finished products. However, this type of conditioning process can be highly unpredictable; even different strains of a microbial species can exhibit notably different growth capabilities and differentially consume or release metabolites that alter beer flavor (8, 17).

Historically, the identification of spoilage microbes relied on the ability to culture a contaminant so that it could be subsequently characterized phenotypically and biochemically (13). More recently, sensitive techniques to detect known spoilage microbes have been developed that employ either image cytometry (24), polymerase chain reactions (PCR) (14), bioluminescence (25), or molecular probing (26). While PCR is excellent for characterizing microbes in postproduction beer and for predicting shelf life, it is unable to detect genes outside those that are targeted, so other potential spoilage microbes go unnoticed. These limitations could largely be overcome using next-generation DNA deep-sequencing to monitor mixed microbial communities because all recoverable DNA can be interrogated and the abundance of nonculturable microbes can also be established (27, 28). Unfortunately, the time and costs associated with deep sequencing are not compatible with routine beer production protocols.

Deep sequencing has been applied to thoroughly evaluate the presence of microbes and particular genes associated with spoilage in an active brewery (12). What emerged from that study were mosaic maps of microbial communities that were influenced both by location and nutrient availability in each brewery station. A main conclusion from that investigation was that repeated exposure to the beer itself was correlated with the abundance of genes that are associated with resistance to iso-alpha acids derived from hops. Thus, in addition to nutrient availability, the chemistry of a given beer, competition or

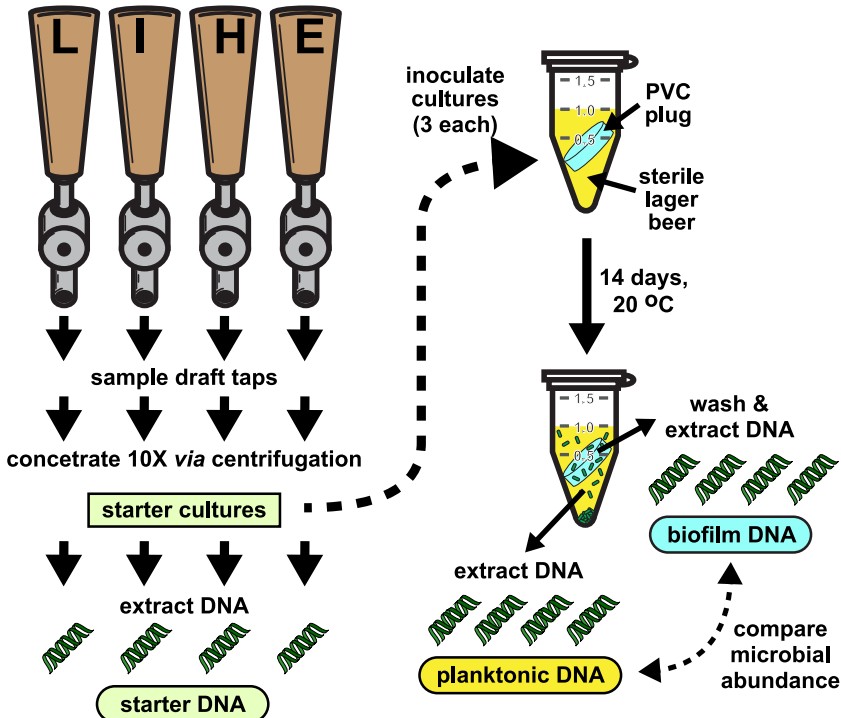

**FIG 1** Beer sampling and biofilm development. Beer samples were collected from four draft taps serving a lager (L), an IPA (I), a hefeweizen (H), or an EPA (E) as the first draws of that day. The microbes in each were concentrated 10-fold using centrifugation to create starter cultures and sampled for DNA extractions. Each culture tube contained sterile lager beer as a growth medium and uniform plugs of draft line plastic prior to inoculation with a starter culture. These cultures were allowed to develop for 2 weeks before extracting DNA from the planktonic and biofilm cells. The DNA in each sample was processed to establish the abundance of different microbes and then compared. Approximately 1 year later, a second sampling was performed from the same taps (which were still serving the same beers) and the experiment was repeated.

predation by other microbes, and evasion from antimicrobial cleaning protocols become key aspects governing a beer spoilage microbial community.

In this study, we recovered the microbial communities from four different beer samples (starters) at two time points from a retail draft system and used DNA deep-sequencing to determine the relative abundances of ribosomal genes in each sample. In addition, each sample was used to inoculate experimental cultures using fresh beer to invoke spoilage in the presence of draft line plastic plugs. The resulting microbial communities of the nonadherent (planktonic) fractions and the stably plastic-associated (biofilm) fractions were subsequently processed and deep-sequenced to establish the relative abundances of the microbes. All together, we detected 119 bacterial and 18 fungal species as contaminants in these draft systems. The samples collected at two different time points yielded different bacterial communities in the same beers. We also identified members of these starter communities that reestablished themselves as members of new biofilms and we were able to identify bacteria that preferred growth in biofilms. Therefore, this study broadens the understanding of beer spoilage beyond controlled brewery settings and sets a foundation for improved retail service education and spoilage monitoring.

## RESULTS

**Establishing a test platform for beer microbiota.** Four beer draft taps at a single retail location were selected for study that delivered a lager (L), an India pale ale (I), a hefeweizen (H), and an extra pale ale (E) (Fig. 1). Aliquots of each draft sample were used to inoculate three replicate cultures. The selected growth medium was the same brand of lager drawn from the tap but sourced from a can to avoid prior microbial

contamination. To ensure sterility, the growth beer was also filter sterilized prior to delivery into the culture tubes. To provide a surface that appropriately represented the beer service lines in this system, the experimental culture tubes also contained uniformly dimensioned plugs of the same type of polyvinyl chloride (PVC) that comprised the retail service lines. These plugs rested at an angle submerged in the culture medium to allow for uninhibited beer exposure and to allow nonbiofilm settling microbes to drift to the tube bottoms. The plug surface area and liquid volume was the same for each replicate. Upon sealing, no additional atmospheric exposures occurred until harvesting. After 1 week, the samples were mixed by vortexing to redistribute the microbes and they were incubated for an additional week. A set of 5 uninoculated medium controls for each study year exhibited no turbidity after incubation and 10 $\mu$l platings of each on malt agar medium yielded no colonies.

**Identifying bacteria and fungi.** The samples were processed to recover total DNA and polymerase chain reactions (PCR) were used to amplify either the hypervariable V3-V4 regions of bacterial 16S ribosomal genes or fungal ITS2 regions between the 5.8S and 28S ribosomal genes (29, 30). These PCR amplicons were then barcoded, pooled, and sequenced using paired-end Illumina technology (Data set S1 and S2) (31). We obtained ITS2 PCR amplicons from each of the year 1 starter samples but were not able to recover amplicons from the year 2 samples, which suggests fungi were in very low abundance. To evaluate the microbial communities, the DNA sequences were computationally processed to identify the source genera and species, along with their relative abundances in the samples (Data set S3-S5). We were unable to obtain bacterial PCR amplicons from the canned lager growth medium, so any genomes present were below the limit of detection and did not substantially contribute to the sequence collections.

Most bacterial genomes contain multiple copies of ribosomal genes and the 16S V3-V4 regions within them may differ in a single organism (32, 33). Moreover, subspecies (strains) of bacteria frequently have the same V3-V4 regions as other members of the species (33, 34). Fungal genomes can have tens or thousands of ribosomal gene copies, which can also be variable within a given species or strain (35–37). Therefore, while the counts of a given sequence and putative species names are informative for comparing community cohorts, they are not directly correlated with cell numbers or strain diversity.

**Microbial diversity in the starter samples.** In the samples collected from the draft taps in the first year, each beer had a different and very rich community structure. In these beers, we identified 164 different V3-V4 sequences that were derived from 98 species of bacteria (Data set S3). Within these samples, we also identified 18 ITS2 fungal sequences and were able to confidently assign 16 species as the sources of them (Data set S4). To gain insight into the dominant community members in these groups, we identified those sequence reads that were present at greater than 1% of the total reads in any given starter sample. A comparison of the community structures revealed that the most dominant bacterial members in the year 1 collection varied substantially between each style of beer, with *Acetobacter*, *Fructilactobacillus*, or *Serratia* as major members (Fig. 2A). Likewise, each starter beer exhibited a notably different fungal community composition (Fig. 2A). A highly abundant *Saccharomyces cerevisiae* sequence (zero-radius operational taxonomic unit [zOTU1]) was present in the hefeweizen sample, which was anticipated because hefeweizens are not filtered prior to service and they are visibly turbid from the brewing yeast. Without additional sequence information on the genomes of these organisms, we are unable to determine if this organism was the same one observed among the other samples because this ITS2 sequence is found in many *S. cerevisiae* strains. The other species are considered to have been beer contaminants.

For the samples collected in the second year, we identified 143 unique bacterial sequences derived from 72 species, most of which were the same as those observed in the year 1 collection (Data set S5). However, the relative abundances of each were markedly different, with *Acetobacter* having dominated all starters (Fig. 2B). These comparisons highlight an important conclusion from this study: although major

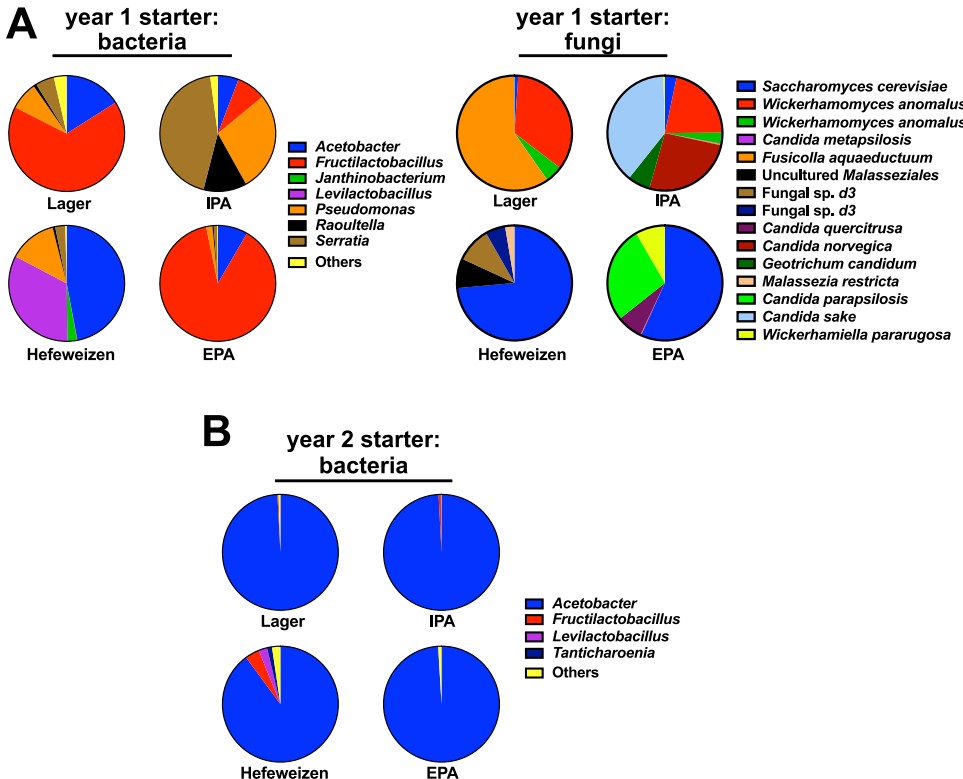

**FIG 2** Bacteria and fungi present in the starter samples. Bacterial V3-V4 and fungal ITS2 hypervariable regions were sequenced and cataloged as zero-radius operational taxonomic units (zOTUs). These sequences were then assigned to source organisms at the genus or species level. (A) Year 1 bacterial and fungi abundances in the starter samples. Pie-charts illustrate the relative read abundances for the indicated organisms. Sequences with read abundances less than 1% of the total were grouped as "others." (B) Bacterial abundances in the year 2 starter samples. Fungal ITS2 PCR amplicons were not recovered from the year 2 starter samples.

community members were similar between the first- and second-year collections, the relative abundances of the bacteria changed dramatically between sampling events. Thus, retail draft line communities can be dynamic and there is no particular pattern of bacterial abundance that could predict which beer they came from.

**Dominant culturable microbes.** We filtered the sequence collection to identify those bacteria that were reproducibly abundant after culturing, which indicates they replicated well in the lager medium. This processing reduced the number of bacterial genera to 31 for the year 1 samples and to 12 for the year 2 samples. These genera were then compared for their evolutionary relatedness within the eubacterial kingdom (Fig. 3). During this analysis, we discovered that approximately half of these genera were predominant in either the biofilm or planktonic culture samples, suggesting that those bacteria had preferred biological niches in our growth experiments (Fig. 3).

Only a few of the yeasts grew well in the culturing experiments and none of those outgrowths exhibited a significant bias between the biofilm and planktonic fractions. In the samples inoculated with the lager-derived microbes, the *S. cerevisiae* and *Wickerhamomyces anomalus* strains grew well in all three culture replicates (Data set S4). The *Fusicolla aquaeductuum* that provided the most sequence reads in this starter sample was not detected in five of the cultured samples and there were only a few reads in a planktonic sample that likely remained from the inoculum without any culture growth. In the cultures inoculated with the IPA starter, *W. anomalus* again grew well along with *Candida metapsilosis*, which was not abundant in this starter. The *Candida sake* that was dominant in this starter did not grow. Of the five prominent yeasts in the hefeweizen starter, only the *S. cerevisiae* grew out in the cultures. Interestingly, a *Brettanomyces* strain that was not detected in this starter was detected in all cultured samples. Its absence in the data for the starter was likely caused by the

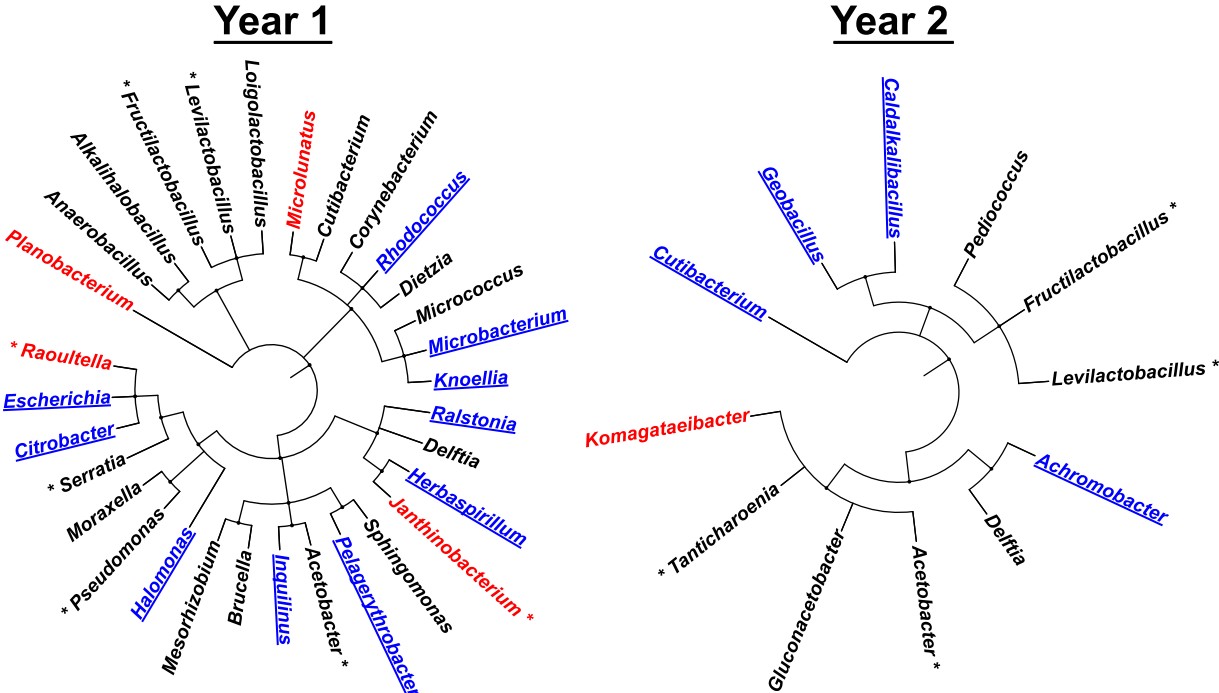

**FIG 3** Diversity of bacterial genera and their preferential growth - taxonomic bushes illustrate the evolutionary diversity of bacteria that reproducibly grew well in the lab cultures, plotted from kingdom to genus for each year. Bacteria with sequence read counts greater than 1% of the total reads in that year are marked with asterisks. Cultured bacteria that were detected predominantly in biofilms are colored blue and underlined; those detected predominantly as planktonic are colored red.

dominance of the *S. cerevisiae* sequence reads in that sample, but there was no correlation between the abundance of *S. cerevisiae* and *Brettanomyces* in the cultures. Finally, of the four dominant yeasts in the EPA starter, only the *S. cerevisiae* grew out in the cultures. These observations indicate that, unlike the recovered bacteria, there was a stark difference between the yeasts that were present in the starter and their capability to grow in the cultures. Overall, the bacterial and fungal community structures that were present in these four starters were not maintained in the lab cultures, even for the cultures grown using the same lager brand.

**Preferences for biofilm or planktonic growth.** The conclusion that some bacteria preferred occupancy in either the biofilm or planktonic communities was derived from a relatively straightforward visual inspection of the read count data for each growth experiment. However, that analysis overlooks bacteria that were abundant in both communities, but whose relative proportions in the biofilm and planktonic communities differed significantly after culturing. Bacteria that show a propensity to grow better in a biofilm relative to other community members may be capable of dominating when biofilms are reestablished after draft line cleaning. To identify bacteria that exhibited such a behavior in our cultures, we applied an analytical method that compares the abundance of a given sequence read relative to a reference sequence that was present in all samples (38). This analysis has an advantage in that it does not require counting the absolute numbers of microbes in a given sample, which is intractable in biofilm studies. We elected to use the *Acetobacter* sequence (zOTU1) as a reference because it was abundant and present in all data sets. We were also able to leverage the outcomes of the replicated cultures to reveal reproducible behaviors.

For this analysis, we first filtered the data sets to only include sequence reads that were present at greater than 0.1% compared to the reference sequence in each of the samples (Data set S3-S5). This filtering strategy avoided a pitfall caused by low abundance sequence reads: small stochastic differences in read counts between replicates can be incorrectly perceived as very large changes in relative abundance. We then calculated how the relative

abundance of a sequence read in the cultured planktonic or biofilm samples changed compared to its abundance in the starter (relative differentials) (38). Taking a $\log_2$ of those values provides an easier interpretation of any 2-fold increase or decrease around zero (zero indicates no change). The relative differentials were averaged across the three replicates and comparisons were made between biofilm and planktonic residency. For example, of the 14 sequence sets that passed the abundance filter in the year 1 lager culture set (a starter, three biofilm, and three planktonic samples in each set), there were seven instances where the relative read abundances were significantly different between the biofilm and planktonic environments (Fig. 4A and B). In this representation, a positive value for the difference between biofilm abundance and planktonic abundance, the delta, means that the bacterium was more prevalent in that biofilm community with respect to the reference.

Strong additional support for this analytical approach came from an interesting discovery we made regarding the five dominant *Fructilactobacillus lindneri* sequences in the year 1 lager experiment (zOTUs 3,4,7,8, and 9), where each was significantly overrepresented in those biofilms by ~50% (Fig. 4B). *F. lindneri* (NCBI RefSeq 3380998) has seven 16S genes and the read counts among the samples for those zOTUs had nearly consistent proportions in all samples of 2:2:1:1:1, respectively. This correlation suggests there was a dominant *F. lindneri* strain with two copies of a 16S gene containing the zOTU3 or zOTU4 sequence, and three 16S genes with each of the others. A similar phenomenon was observed with *Loigolactobacillus backii*, where its five 16S genes appeared in a 3:1:1 ratio (zOTUs 14, 18, and 19), indicating that three 16S sequences were identical and the remaining two were different. An *L. backii* representative genome (NCBI RefSeq 3405468) also shows a 3:1:1 ratio of V3-V4 region sequences. In contrast to *F. lindneri*, the log-ratio comparisons between culture and starter revealed that these *L. backii* sequences consistently had positive values (Fig. 4A), although insignificant deltas between the biofilm and planktonic fractions (Fig. 4B). These results overall indicate that the abundances of 16S genes from these bacteria were changing in consistent proportions regardless of their copy number, which would be a requirement if they were members of the same genome. Moreover, this type of analysis may be useful in future studies for teasing out the number of strains within samples that have similar or shared gene sequences.

We performed the filtering and delta analysis for the seven other culturing experiments (three other beers from year 1 and four from year 2) and identified five sequences that also exhibited significant deviations between the biofilm and planktonic samples (Fig. 4C) (Data set S3 and S5). Interestingly, the *F. lindneri* zOTU3 and zOTU4 sequences were also overrepresented in the year 2 lager biofilms (by ~30-fold) and the corresponding zOTUs 7, 8, and 9 were again present in the same ratios as was observed in the year 1 experiment. Unfortunately, the read counts of the latter three were too low in the starter sample to survive our filtering protocol. Nonetheless, this organism grew well in the lager, and it became proportionally more abundant in the biofilms.

Overall, the microbial communities in each draft line became restructured over the course of a year and the dominant members changed, which indicates that the chemistry of the beers themselves was not fully defining those community structures. These observations highlight an important value in 'field monitoring' microbial communities in different industrial or clinical settings because single snapshots of diversity in complex microbial communities neither reflect their history nor predict their futures.

## DISCUSSION

We have provided a comprehensive survey of microbial communities that can inhabit retail draft beers and determined their ability to establish new communities in a controlled setting. In the cases of the two experiments with lager-derived microbes, the organisms were presented with the same nutrients as their source environment, whereas the other communities were forced to adapt from their source beers into the lager. At first glance, it may seem surprising that the communities became so different from their starters when

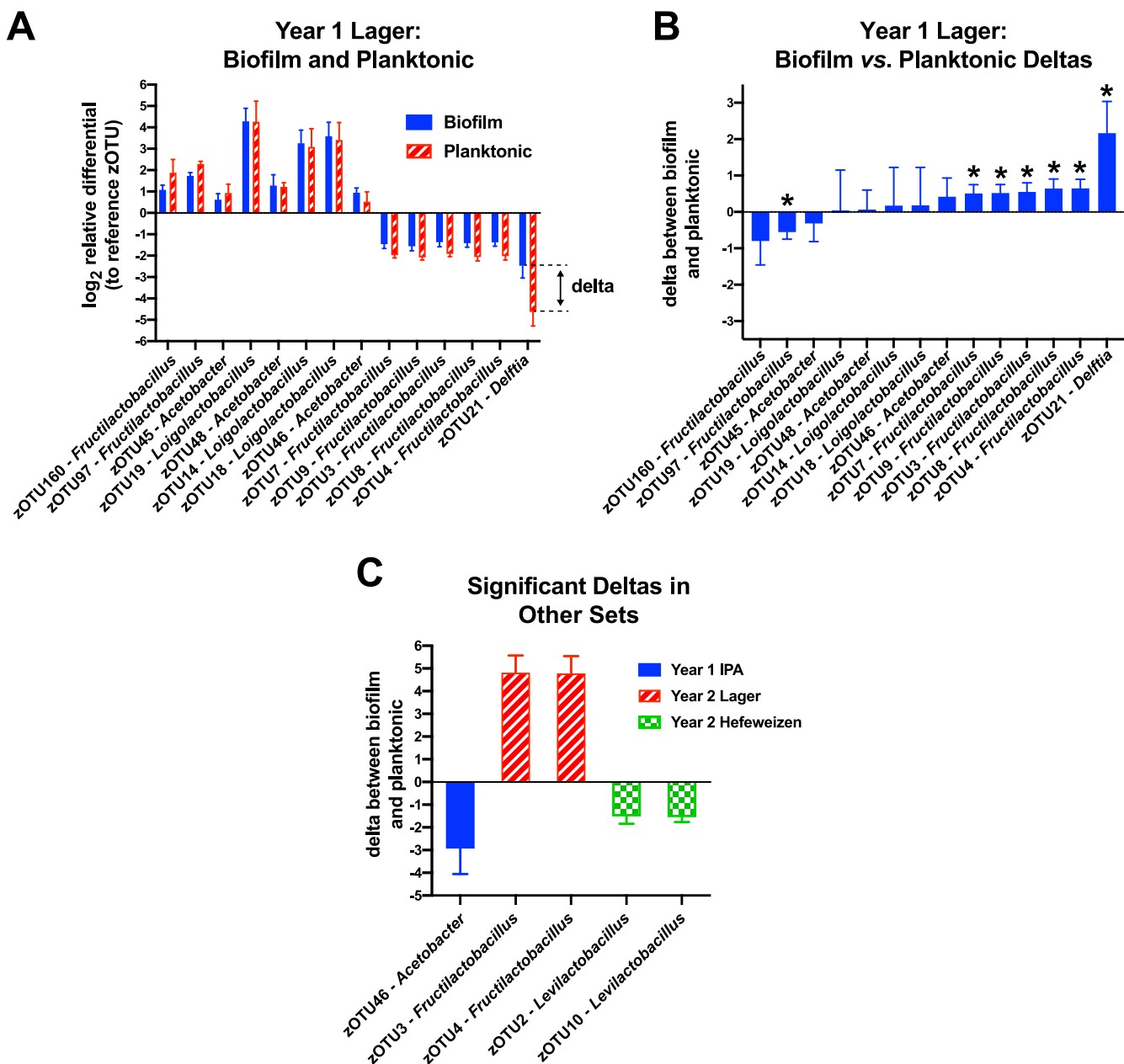

**FIG 4** Evaluating bacterial biofilm preferences. The sequence read counts in each sample were used to calculate relative abundances with respect to a common reference sequence in each sample (zOTU1). Those ratios were then used to establish changes in their relative abundances in the incubated biofilm or planktonic communities compared to their abundances in the starter samples. (A) Bar plots of the $\log_2$ transforms of the relative differentials for bacteria in the incubated lager cultures (a 2-fold change is one unit on the ordinate axis). Error bars indicate the standard deviations between the three culture replicates. The 'delta' is the difference between the biofilm change and the planktonic change. (B) Bar plots of the deltas, with negative values indicating a preference for the planktonic niche and positive values for the biofilms in the cultured samples. Delta values from pairs that had significant differences between the biofilm and planktonic groups are marked with asterisks (t test P values <0.05). (C) Five additional significant biofilm deltas observed among the other seven culturing experiments.

cultured. However, the starter communities themselves were recovered from environments that were already in flux: because they were recovered as the first draft pours of the day from unflushed lines, they were in the process of consuming leftover resources and they had not been supplied fresh nutrients or inhibitors for ~12 h or more, depending on the last time beer was pulled completely through each system.

Another variable to consider is that the starters were planktonic communities, not draft line biofilms. However, for most of the observed microbes, there was no significant bias in their ability to become members of a new biofilm. This finding suggests

that the recovered planktonic communities may have been a fair reflection of any biofilms that were in those draft systems. More studies will be needed to determine how an established biofilm community can populate uncontaminated beer, which we suspect is a major driving force for postproduction spoilage. In addition, the lab culturing occurred over 2 weeks and the dynamics of the communities over that time are unknown. The choice of a long culturing stage was motivated by our experiences with isolated cultures of *Acetobacter* and *Lactobacillus*, which can take several days or more to reach saturation in established culture media or sterile beer. Nonetheless, we were able to evaluate outgrowth and to identify bacteria with preferences for biofilm and planktonic growth.

The chemical properties of the source beers should also be considered because they are expected to shape the microbial communities that feed on them. Hops is used in beer production to impart desirable flavors and aromas, but also to suppress bacterial growth (11). Therefore, we expected to see substantially less bacterial diversity in the higher hop content beers (EPA and IPA), which was not the case. In addition, the IPA contained twice the alcohol content of the lager. A reciprocal experiment (placing different starter communities in fresh IPA and monitoring the outgrowth) will be required to assess the impact of those chemicals on community dynamics. Most studies on antibiotic efficacy are conducted using a single organism, which does not allow for intermicrobial metabolisms to be evaluated. Mixed bacterial species can connect their cell bodies to each other using nanotubes that allow for the sharing of enzymes and chemicals, which greatly expands the metabolic capacity of the community as a whole (39). If such sharing occurs in beer biofilms, then the observed communities may reflect a global survival strategy that is selected for resistance to beer ingredients and cleaning chemicals.

It is not surprising that many of the predominant members of the retail draft line communities are well known for causing beer spoilage in breweries, such as LABs and AABs. LABs are considered to be one of the dominant contaminant microbes in breweries and they can contain genes that render them hop-resistant, so they can persist in beers with high hop extract content (15, 40, 41). In a comprehensive survey of brewery microbes found at different brewing stages (12), it was revealed that bacteria in the family *Lactobacillaceae* had a high relative abundance in beer samples, but much less occupancy elsewhere in the brewery. This disparity likely reflects the fact that these bacteria commonly prefer to metabolize as anaerobes, and they grow poorly or not at all when exposed to the open atmosphere (42). The methodology of that brewery study limited the taxonomic assignments to the family level; however, in our analysis we could confidently assign a species to strains of *Fructilactobacillus lindneri*, *Levilactobacillus brevis*, *Loigolactobacillus backii*, and *Pediococcus damnosus*. Although these microbes routinely contaminate beers and are frequently consumed, they are not considered to be human pathogens (43, 44).

A recent investigation of the growth properties of *F. lindneri* indicated that this bacterium can enter a "viable but nonculturable" (VBNC) state at low temperatures and that it requires anaerobic conditions for robust growth (45). Therefore, it is missed by routine colony screening for contamination and can persist undetected in refrigerated beers for long periods of time. That study also revealed that *F. lindneri* cells can produce high levels lactic acid, acetic acid, and diacetyl as waste metabolites, even from the VBNC state. The detection of this bacterium in all of our beer samples and the robust growth the lager cultures indicates that it may be a major contributor to the off flavors encountered in the starter beer collection. Our finding that it also prefers to reside in the biofilm niche suggests it may be one of the community members that evades cleaning protocols. A convenient and low-cost technology to monitor the presence of this microbe in retail settings would be useful for postproduction quality control.

The *Levilactobacillus brevis* strains identified in our study also belong to another known beer spoilage group (46, 47). They were abundant in the hefeweizen samples, but at least one strain was detected in each starter sample. Some strains of this bacterium have been well characterized because they are considered probiotic and are used

in the production of yogurts, kimchi, sourdoughs, and other fermented foods (48–52). In addition to producing lactic acid, these bacteria tend to secrete copious amounts of exopolysaccharides that cause slimy textures and increased viscosity (53, 54). Thus, not only does their presence sour beers, but abundant growth causes unpleasant mouthfeel.

The *Acetobacter* strain that was the most prominent contaminant in our samples did not receive a species assignment for this study because although its V3-V4 sequence matches *A. farinalis*, it is over 99% similar to several other strains, including *A. orleanensis*, *A. persici*, and *A. cerevisiae*. Most of the characterized members of these species have been recovered from spoiled beers (55), and *A. farinalis* from fermented rice flour (56). As the genus name suggests, these bacteria produce acetic acid form a variety of sugars and ethanol, but they can also consume ethanol, acetic acid, and lactic acid as energy sources if oxygen is present (57, 58). The dominance of *Acetobacter* in each of the year 2 samples suggests there may have been an alteration in the maintenance of the draft lines that allowed more oxygen exposure. Limiting oxygen exposure is a well-established beer brewing mandate and we suggest limiting oxygen exposure during draft line servicing.

The original sources of the microbial contaminants are unknown, but observation of the same genera and species in each of the beers (brewed in different facilities) suggests that they are, for the most part, residents of this retail setting and not continuously supplied from the kegs themselves. Also, the communities changed in all four beers between the two sampling dates, so the likelihood that each brewery encountered the same shifts in microbial contamination is unlikely. Having access to source beers would be one way to test this conclusion, but depressurizing and accessing fresh kegs for sampling prior to this study was not logistically feasible. In addition, an initial draft sample would have passed through a contaminated line already. Although the presented data were derived from cultures obtained at a single retail location, we have tested draft beer samples from seven other locations by plating them on malt agar containing cycloheximide (to inhibit yeast) and each contained substantial bacterial contamination. Some retailers flush their service lines before the first pours of the day, while others do not. In either case, having better control of in-line spoilage would help maintain beer quality.

So, what is a retailer to do? One obvious approach is to increase or change line cleaning events. However, we noted that when beer draft lines are cleaned, a common methodology is to remove the keg coupler, connect the line to a keg with cleaning solution, flush the line, and then reconnect the coupler back onto the same keg. If a beer in a keg or its valve was already contaminated, no amount of line cleaning will prevent regrowth of a spoilage community. Also, if multiple lines are cleaned using the same cleaning keg, cross contamination between lines is very likely. Perhaps the couplers and keg valves could be disinfected during keg changes and line cleaning. On top of this, there is no way to prevent contamination at the tap end: it is fully exposed to the far-from-aseptic bar environment.

Another consideration is the draft line material. Occasionally replacing lines is recommended (59), but there is no formal guideline on when, what specific material to use, or how to diagnose a serious biofilm problem. Cost of line replacement and downtime are other economic considerations. From this study, we can suggest that perhaps the draft beers be monitored for *Acetobacter* because they were common in all samples. We have recently isolated strains of *Acetobacter* from this draft system and are evaluating their ability to form biofilms on various materials. What we have so-far discovered is that they behave differently and produce substantially more biofilm than a reference *A. cerevisiae* strain we obtained from a stock center (DSM 2324, Leibniz Institute). That reference strain was recovered many decades ago from a brewery in Germany, long before modern draft line plastics were available. We suspect that the continued selection for bacteria and wild yeast to survive on a given surface during cleaning episodes has created collections of microbes that are tailored to be resilient

in each environment. This is a completely analogous situation to the problems caused by microbial biofilms in medical and industrial settings. For these reasons, additional biofilm growth and cleaning research is warranted. As indicated earlier, taking advantage of the 16S gene copy numbers in different bacteria may be useful in teasing out whether or not there are multiple strains and for validating conclusions drawn from the sequence read abundances.

This project was inspired by anecdotal observations of dramatic taste and odor changes in draft beers served at several local restaurants, a phenomenon we affectionately refer to as a 'bowling alley' taste. After preliminary plating and Sanger sequencing experiments, we recognized that the diversity of microbes was far greater than our expectation of a few dominant contaminants. This research was intended not only to expand our understanding of microbial community behaviors, but also to provide an impetus for brewers to consider the downstream influences on the perception of their beer quality: a customer trying a new beer will be offput by spoilage metabolites and probably never realize that the flavor is substantially distorted. Likewise, bartenders are rarely trained (or allowed) to evaluate service beers, so they may never realize there is a problem with the product. Another aspect of contaminated beer is the potential for health impact. It is known that humans obtain their microbial flora from environmental exposure and that gut microbiology is dictated by diet (60). Therefore, this study provides a motivation for a formal study to be conducted to establish relationships between beer spoilage microbes, resident gut flora, and human physiology.

## MATERIALS AND METHODS

**Sample collection and culturing.** Samples were collected from retail drafts that were supplied from kegs maintained at 4°C. The drafts were supplied using 1/2 in. clear PVC tubing, which exited the cooler and ran ~25 ft ambiently to refrigerated draft service heads. The tubing was approximately 10 years old and exhibited turbidity on the interior surfaces. This retail location does not flush beer lines prior to daily service; therefore, the beers were collected as the first draws of the day without line flushing. Approximately 200 ml of draft samples of four beers from different breweries were collected into sterile cups: an American lager ("L," unpasteurized, 4.5% ABV, ~15 IBU), an India Pale Ale ("I," unpasteurized, 9% ABV, ~90 IBU) a hefeweizen ("H," unpasteurized, 5.3% ABV ~18 IBU), and an extra pale ("E," unpasteurized, 5.7% ABV, ~40 IBU). These values were taken from the manufacturer websites, which are not referenced to maintain anonymity. After mixing by swirling, 50 ml was transferred to sterile conical tubes and placed on ice. Approximately 45 min later, each tube was centrifuged for 30 min at 3,500 RCF at 4°C and 45 ml of the cleared supernatants was removed by aspiration. The pellets were resuspended in the residual 5 ml, creating 10× "starter" stocks. 1 ml aliquots of each sample were then prepared: one was frozen at −80°C and the others were kept on ice to serve as inoculation sources that day. This process was repeated the following year from the same taps.

Solid substrates for biofilm development were formed by hand-stamp punching 0.25 in. plugs from 3/8 in. flexible PVC (Vinyl-Flex NFS-61, Advanced Technology Products, Milford Center, Ohio) and collected into a clean glass beaker. In a sterile laminar flow hood, the plugs were then submerged in a peracetic acid sterilizing solution (SporGon, Decon Labs Inc., King of Prussia, PA) for 3 h, and then rinsed four times with 0.22 $\mu$m filter-sterilized HPLC-grade $H_2O$. After the last wash, the residual water was drained, and each plug was transferred using sterile tweezers into sterile 1.7 ml microfuge tubes and stored until use.

Sterile growth medium was prepared by 0.22 $\mu$m filtering a commercial canned lager that was the same as the sampled American lager, except it had been pasteurized prior to canning. 1 ml aliquots were then aseptically transferred to the tubes containing the PVC pellets. A set of 5 uninoculated tubes was set aside as sterility controls for each year's study and triplicate experimental tubes were inoculated with 10 $\mu$l of the 10× microbe starter stock and mixed by vortexing. The culture tubes were then placed in a 20°C incubator for 2 weeks with an additional vortex mixing event after the first week.

**DNA extraction.** Culture tubes were vortexed briefly to resuspend settled cells and a 500 $\mu$l aliquot was set aside as the planktonic cell fractions. The PVC plugs were then transferred with sterile tweezers to a microfuge tube containing 1 ml of 0.22 $\mu$m filter-sterilized HPLC-grade $H_2O$ and vortexed to remove nonadherent cells. This washing step was repeated two more times.

Bead ablation tubes were prepared by adding 100 $\mu$l of 0.1 mm zirconia beads (Research Products International, Mount Prospect, IL) to sterile 2 ml screw cap microcentrifuge tubes. 500 $\mu$l of a denaturing "extraction buffer" (5.5 M guanidinium thiocyanate, 100 mM potassium acetate, pH 5.5) was added to the beads before adding 200 $\mu$l of either planktonic cells or 200 $\mu$l of sterile HPLC-grade water and a PVC plug. The tubes were then agitated twice using a FastPrep-24 lysis 5G instrument (MP Biomedicals, Irvine, CA) using the "Escherischia coli cells" setting. After disruption, cell debris and beads were collected by centrifugation at 14,000 RCF for 5 min and 500 $\mu$l of each cleared supernatant was transferred to a clean tube. 200 $\mu$l of isopropanol was added and mixed by vortexing and the solution transferred to a DNA binding silica spin column (EconoSpin, Epoch Life Science, Missouri City, TX). After passing the solution through the column twice, the column was washed with 300 $\mu$l of extraction buffer, followed by three washes with "column wash buffer" (80% ethanol, 10 mM Tris-Cl, 0.1 mM EDTA, pH 8.0). The

columns were then dried by centrifugation and the samples eluted in 50 $\mu$l of "DNA buffer" (5 mM Tris-Cl, 0.1 mM EDTA, pH 8.0). DNA samples were stored at $-20°C$.

**Library preparation and sequencing.** Bacterial 16S V3-V4 regions were amplified by PCR using universal primers derived from DBact-0341-b-S-17 and S-d-Bact-0785-a-A-21 (29) containing unique adapters for the Nextera XT indexing kit according to the manufacturer's instructions (Illumina Inc., San Diego, CA). Fungal ITS2 regions were amplified using separate adapter primers derived from IST3_KYO1 and IST4_KYO1 (30). After adding unique indices, each sample's DNA concentration was determined using the Quant-iT PicoGreen kit (Molecular Probes, Inc., Eugene, OR) and equal mass portions of each were pooled prior to sequencing.

The amplicon pools were paired-end sequenced (250 cycles) using the MiSeq platform (Illumina Inc.) at either the Interdisciplinary Center for Biotechnology Research (year 1 data set, University of Florida, Gainesville, FL) or the UCF Genomics & Bioinformatics Cluster (year 2 data set, University of Central Florida, Orlando, FL).

**Sequence processing and bioinformatics.** Trimmomatic was used to remove primer sequences and to filter out reads less than 150 bases long as well as reads with Phred quality (Q) scores less than 25 using a sliding window of 4 bases (61). VSEARCH was used to merge the forward and reverse sequences to generate the complete V3-V4 regions with a minimum total merged sequence length of 200 bases, minimum overlap length of 50 bases, and a maximum of 5 allowed mismatches across the alignment (62). Following merging, sequences with an expected error rate >1 were discarded. The remaining sequences were then dereplicated while counting the number of each unique sequence.

VSEARCH was used to denoise reads, remove chimeras, and to generate zOTUs. Parameters for zOTU clustering included occurrence of a minimum of 50 unique reads within a 99% identity threshold. All error-filtered merged sequences were then mapped to zOTU sequences based on 99% alignment to generate a table containing sequence read counts per zOTU for each sample.

Taxonomic classification of zOTU sequences was carried out using the Bayesian Lowest Common Ancestry (BLCA) software, which employs full-length query-hit alignment scores to generate a weighted probability for taxonomic allocation (63). For bacterial classification, the NCBI 16S rRNA database was used (updated 06/24/2021) and for fungal classification the UNITE v6 database was configured for usage with BLCA (64). Classification was declared based on the lowest common ancestor with a cumulative posterior probability $\geq$ 80%. Fungal zOTUs were additionally manually compared to the NCBI's nonredundant eukaryotic database using BLAST (65, 66).

The percent abundance of each zOTU was determined in each sample's data set and used to calculate ratios relative to a common reference (zOTU1, which was present in all bacterial samples) as previously described (38). Finally, the $\log_2$ transforms of these reference frames were used to evaluate relative changes in the bacterial communities during culturing and to compare the relative abundances in biofilms and planktonic samples.

**Data analysis and graphics.** Taxonomy bushes were generated using the Interactive Tree of Life (iTOL) (67). Data were sorted and processed using Excel (Microsoft, Redmond, WA). Data were plotted using Prism (GraphPad, San Diego, CA) and figures were generated using Illustrator (Adobe Inc., San Jose, CA).

**Data availability.** Raw Illumina sequences are available on request to the corresponding author. Processed zOTU sequences and data analytics are components of the associated Supporting Information.

## SUPPLEMENTAL MATERIAL

Supplemental material is available online only.
**SUPPLEMENTAL FILE 1**, XLSX file, 0.04 MB.
**SUPPLEMENTAL FILE 2**, XLSX file, 0.01 MB.
**SUPPLEMENTAL FILE 3**, XLSX file, 0.5 MB.
**SUPPLEMENTAL FILE 4**, XLSX file, 0.1 MB.
**SUPPLEMENTAL FILE 5**, XLSX file, 0.4 MB.

## ACKNOWLEDGMENTS

This work was partially supported by NIH grant 1R01GM118896 and a UCF "What's Next" Quality Enhancement Plan grant. We thank Anna Ward and Laurie Agosto for their technical assistance, as well as the following UCF undergraduates who participated in this project as members of the UCF Applied Industrial Microbiology program: Cesar Ballesteros, Taylor Croteau, Mahammad Gardashli, Haley Hardin, Christopher Hawkins, Sarah Kampiyil, Ashley Lima-Acosta, Jordan Palmer, Vanessa Parra-Gonzalez, Tristan Tran, Michael Tucker, Paola Andrade, Erick Bardalez, Danielle Beetler, Amy Freiberg, Chanelle Hunter, Hayden Kim, Imani Pascoe, Dana Ramsey, Keith Taylor, Evie Vincent, and Emily Wilson.

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
