## [Reviewer comments · Microbiology Spectrum]

Microbiology Spectrum

Microbial communities in retail draft beers and the biofilms they produce

Nikhil Bose, Daniel Auvil, Erica Moore, and Sean Moore

Corresponding Author(s): Sean Moore, University of Central Florida

Review Timeline:

Submission Date:	September 6, 2021
Editorial Decision:	October 2, 2021
Revision Received:	October 25, 2021
Accepted:	November 3, 2021

Editor: Jeffrey Gralnick

Reviewer(s): Disclosure of reviewer identity is with reference to reviewer comments included in decision letter(s). The following individuals involved in review of your submission have agreed to reveal their identity: Caleb Levar (Reviewer #2)

Transaction Report:

DOI: <https://doi.org/10.1128/Spectrum.01404-21>

October 2, 2021

Dr. Sean D. Moore
University of Central Florida
Burnett School of Biomedical Sciences
4110 Libra Drive
BMS 125
Orlando, FL 32816

Re: Spectrum01404-21 (Microbial communities in retail draft beers and the biofilms they produce)

Dear Dr. Sean D. Moore:

Thank you for submitting your manuscript to Microbiology Spectrum. While the reviewers were both enthusiastic about the work overall, a number of issues have been raised which will improve the accuracy and clarity of the manuscript. In particular, please pay close attention to where the reviewers question if the data shown support the claims made. In most (or all) of these cases additional experimentation may not be necessary - adjustments to the text should be sufficient. This is a really cool paper and I am looking forward to the revision!

When submitting the revised version of your paper, please provide (1) point-by-point responses to the issues raised by the reviewers as file type "Response to Reviewers," not in your cover letter, and (2) a PDF file that indicates the changes from the original submission (by highlighting or underlining the changes) as file type "Marked Up Manuscript - For Review Only". Please use this link to submit your revised manuscript - we strongly recommend that you submit your paper within the next 60 days or reach out to me. Detailed information on submitting your revised paper are below.

Link Not Available

Sincerely,

Jeffrey Gralnick

Journals Department
Reviewer comments:

Reviewer #1 (Comments for the Author):

Thank you for your contribution to the field and participation in peer review and scientific dissemination. My attached document has many comments and suggestions to hopefully improve the manuscript.

Reviewer #2 (Comments for the Author):

Major issues:

In the abstract, the authors make the claim that the methods used here serve as "as starting point for efficient monitoring of beer

spoilage..." (Line 33-34) and suggest that their data regarding biofilm growth will help define more appropriate draft system cleaning protocols. There are a number of issues with these claims. First, draft beer spoilage is not necessarily defined by the presence of viable spoilage organisms. Instead, off flavor and aroma characteristics are the most important trait that would indicate altered cleaning regimes. The authors should indicate if the samples used had off flavor characteristics, either by sensory analysis (blinded triangle testing, as an example) or through more quantitative means (Such as HPLC, etc). Second, culturing samples is not overly useful to an establishment or brewery, as it provides a retrospective analysis of what was present at some point in the past-by the time this protocol has been followed, the contamination problem could have gotten worse to the point that more serious cleaning protocols are required. Last, typical draft cleaning protocols are expressly designed to disrupt biofilms and eliminate bacterial and fungal contaminants. For example, cycles of acidic and basic solutions are pumped through the lines using techniques to introduce turbulent flow intended to disrupt organic buildup/biofilm as well as any mineral scale that may be present. If the authors wish to make the claim in line 33-34, a more thorough discussion of current draft beer monitoring, cleaning, and sanitizing protocols must be included in this manuscript, along with a discussion of how their outgrowth protocol will specifically be useful to the various businesses within the draft beer supply chain.

One could argue that this paper is more specifically assessing the line cleaning regimen of the particular institution from which the samples were taken, rather than a more general assessment of beer spoilage organisms. Is it known how (or if) this establishment was cleaning their draft lines, and if said protocols were in line with industry standards? How long had it been since the last cleaning? Is it known if line cleaning protocols changed between year 1 and year 2, or if the lines were cleaned the same time ago or each sampling?

Why was the lager beer chosen for the growth medium? How might the relative levels of inhibitory compounds (EtOH, hop compounds, etc) between the different beers alter the outgrowth of microbes from the different samples? Could the changes in outgrowth observed be explained by this? A discussion of the similarities and differences between lager-lager outgrowth and (for example) ipa-lager outgrowth should be included

Growth temperature vs draft line temperature: The authors chose an incubation temperature of 25 degrees C. Most draft systems operate at ~4 degrees C (or less) to ensure CO₂ does not break out of solution and to limit contaminant growth. The authors note that the diversity of the outgrown samples changed dramatically from the initial "starter". How much of this difference might simply be explained by growth rate differences at the increased temperature chosen for culturing?

Minor Issues:

The authors make the point that growth as a biofilm is important to the persistence of beer spoilage organisms in draft systems. However, the authors did not directly sample surface adhered/biofilm organisms from the draft lines, instead resorting to liquid samples. This likely will bias the initial abundance of organisms as well as the resulting outgrowth. A discussion of these biases should be included

The authors indicate that they sampled from the first pour of the day, after beer sat stagnant in the draft lines overnight. Typically, this first pour (Or more) is discarded until this lines have been purged to ensure that stagnant beer with greater potential for off flavor characteristics (due to spoilage, temperature differences, co₂ loss, etc). As such, the sample used for these experiments would not be representative of product served during service. A discussion of why this particular sample was used should be included.

Discrepancy between text and figure: In figure 1B an incubation temperature of 20 degrees C is shown. In the text, line 448 an incubation temperature of 25 degrees C is shown.

Staff Comments:

Preparing Revision Guidelines

Please return the manuscript within 60 days; if you cannot complete the modification within this time period, please contact me. If you do not wish to modify the manuscript and prefer to submit it to another journal, please notify me of your decision immediately so that the manuscript may be formally withdrawn from consideration by Microbiology Spectrum.

Thank you for permissions to peer review this study

Review of title:

Title accurately reflects the manuscript

Review of abstract and importance:

Abstract accurately reflects the manuscript and presents valid impact.

Importance section utilizes easily understood verbiage to demonstrate the importance of the study.

Review of manuscript at large:

This research study presents applicable microbiological analysis of an often-neglected area of beverage studies. The author is accurate that there has been a discrepancy in quality assurance/control investigations and peer-reviewed studies addressing draft lines compared to other stages of beer production. Potential draft-line contamination is an under-tested but vitally important part of consumer exposure. The study combines DNA sequencing surveys with laboratory incubations well. The study utilizes deep sequencing appropriately and rightly avoids overly conclusive causations/associations/comparisons. This proper survey of the microbes present and their relative abundancies shows 'rampant' contamination, but high 'flux' and potential stochasticity in draft dispensed beers. Laboratory microcosms were set up to provide multiple niches for microbial propagation to survey microbes that may thrive in the draft system as planktonic or within a biofilm. This presented survey of information, including key prevalent microbes, may become important in future development of quality procedure development.

Review of text details:

Larger critical edits:

- The temperature utilized for incubation must be uniformly referred to throughout the manuscript. Whether it was either 20°C or 25°C or a range 20-25°C, it should be stated uniformly throughout.
- The sterile controls are mentioned in the methods (line ~446) but are not mentioned later in the manuscript. Their results are vitally important and are worth stating to validate the study. Were they analyzed, in what way, what was the result?
- Details about beer 'habitat' and production methods are necessary to appreciate the study more and to have more potential follow-ups. This crucial information should be mentioned in the sample collection and culturing section of the methods. Examples of desired information (if known): Are beer(s) are filtered/centrifuged/pasteurized/barrel aged or not? What is the approximated IBU of the beers? What is the approximate ABV of the beers? Did any of them come from the same production facility? Is there anything known about the yeast(s) used in their productions? Any other researched details are worth noting.
- Details about beer dispensing 'environment' should be elaborated to appreciate the study more. This crucial information should be mentioned in the sample collection and culturing section of the methods. Examples of desired information (if known and willing

to be divulged): What was the make-up of the lines: plastic type and diameter? Is this considered a long or short draft/approximate length of lines? What was the approximate temperature of the beer and/or keg storage? What was the standard cleaning procedure and schedule for such? What was the night storage within the lines? Were the lines allowed to flow or not prior to sample collection?

- The additional experiment that would directly strengthen the manuscript significantly is a DNA analysis of the beer without exposure to the draft lines. If this is possible for any of the beers it should be done; if it is not logistically feasible it should be mentioned as such.

Minor critical edits:

- Citations warranted:
 - line 75 citation(s) warranted to support industry's revisitation
 - line 80 citation(s) warranted to support the primary source of spoilage
 - line 380 citation(s) warranted, even if unpublished, to support the statement that "we have not identified any..."
- Word choice/grammar/elaborative suggestions:
 - Line 59 "go to waste" (word choice, overtly strong – ...are diminished? ...are significantly challenged?)
 - Line 76 "is to develop create products" (grammar – develop creative products? Develop and create products?)
 - Line 101 "abundance of a particular microbe's genome" (word choice, whole genome sequencing vs. amplicon sequencing – abundance of a sequence stretch of a particular microbe's genome? Abundance of a particular microbe's genomic region? Abundance of a particular microbe's genomic taxonomic marker?)
 - Line 121 "relative abundances of the microbial genomes" (word choice, whole genome sequencing vs. amplicon sequencing – relative abundances of the microbial taxonomic genes?)
 - Line 123 "nutrient-rich growth environment" (word choice, as finished beer isn't generally considered nutrient-rich – potentially habitable growth environment? An appropriate proxy environment?)
 - Line 148, 448, etc "mixed" (elaborate how the cultures were mixed, every time it is mentioned – mixed via inversion? Mixed via vortexing?....?)
 - Line 320 "first draft pours of the day" this elaboration isn't just a discussion point, this is a notable methodical point. It is worth mentioning elsewhere as well; for example, it is worth mentioning in methods line ~426.
 - Line 325 "long culturing stage" (word choice, "long" is relative and two weeks is short to many investigations – two-week culturing stage?)
 - Line 340 "these bacteria are usually anaerobes" (word choice, as these are primarily facultative organisms and the environment impacts their current metabolism – commonly metabolize as anaerobes?)
 - Line 345 "In another study, Although these" (grammar and confusion)
 - Line 354 "indicates that it is likely a" (word choice – it is maybe/possibly?)

- Line 399 “A. cerevisiae strain we obtained” (elaborate with strain number, or some identifier)
- Line 427 the beer type abbreviators, L, I, H should be reiterated

Review of figures

- Figure representation:
 - I think the figures being regrouped into DNA extraction type would help further understanding and flow. Figure 1, methods, should remain figure 1 (highly suggested elaborations/completions are below). Figure 2 should be of starter sample DNA extractions, to establish a wholistic baseline; this would compile the current figure 2B and figure 4. Although probably motivated by logical domain categorization, as is, the current figure 4 (of starter DNA fractions) reads as a somewhat disorienting afterthought. The current figure 2A should be its own figure, as figure 3, as it starts to compare starter to incubated DNA assemblages. The current figure 3 would then become figure 4 that compares biofilm and planktonic fractions from incubated DNA fractions. This new figure order would flow nicely as the methodic figure 1 lays them out. The labeling of DNA fractions should be well established within figure 1 and verbiage should be reused exactly as figure headers/titles in the later figures. (e.g. starter DNA, incubated DNA).
- Figure 1 and legend:
 - The last sentence of legend, lines 544-546, applies to both A and B of the figure so should be present prior to the subpart descriptions, if kept as distinct subparts of the same figure.
 - A combined figure showing the connectivity of the ‘field sampling’ and the ‘lab experiments’ would accomplish the methodic story better than distinct subfigures.
 - If a methods figure is being utilized, more details should be included so that the details of the whole procedure can be understood at a glance and be referenced to when digesting the paper as a whole. This could aid the understanding of the future figures. A key missing aspect is the depiction of the 3 different sources of DNA material, which are later compared (starter DNA, incubated planktonic DNA, and incubated biofilm DNA).
 - Extract DNA → extract starter DNA
 - Sterile beer → sterile lager beer
 - Depict some details from within your methods section alongside the arrows (e.g. 10x concentration via centrifugation)
 - An arrow/arrows should be drawn from the bottom of the currently part A of figure (start cultures) to the initial culture
 - The figure states 20°C incubation, but the text of the manuscript states 25°C – fix for uniformity, whichever is accurate
 - Continue depicting the rest of the manuscript’s experiments; it currently ends with incubation and doesn’t explain the crucial DNA subfractions that comes from them.

- Draw two arrows from the mature culture tube and depict planktonic and biofilm fractions, and then depict DNA extraction of each
- Figure 2 and legend
 - Lines 554 that defines the asterisk, should be elaborated; >1% of what? “>1% of total reads” “>1% of total of starter DNA” “>1% of total cultured DNA” or similar verbiage
 - 2A, bushes: the color variation of black, blue, red are indistinguishable on grey scale (and impaired vision); underlines or denotation of some kind could help accessibility
 - The titles of figure 2B should state ‘starter’ along with the year sampled
- Figure 3 and legend:
 - 3A & 3C, bar graphs: the color variation of green, blue, red are indistinguishable on grey scale (and impaired vision); patterns or denotation of some kind could help accessibility
 - Refer to incubated DNA appropriately
- Figure 4 and legend:
 - Line 574, “fungi present in the samples” – should be elaborated to “present in the starter samples”
 - The lines of text from ~267 should be reiterated in the figure legend to help the reader understand why there isn’t a chart for year 2
 - The sentence currently in figure 4 legend on line 578, “the dominant...” Is a discussion point and hypothesis and not exactly appropriate for a figure legend.
 - The titles of figure 4 should state ‘starter’ along with the year sampled

Dear reviewers,

We revised the manuscript to address concerns that were presented. In response to a suggestion from Reviewer 1, we restructured the manuscript to follow the workflow presented in the revised Fig. 1.

Reviewer 1

Review of title:

Title accurately reflects the manuscript

Review of abstract and importance:

Abstract accurately reflects the manuscript and presents valid impact.

Importance section utilizes easily understood verbiage to demonstrate the importance of the study.

Review of manuscript at large:

This research study presents applicable microbiological analysis of an often-neglected area of beverage studies. The author is accurate that there has been a discrepancy in quality assurance/control investigations and peer-reviewed studies addressing draft lines compared to other stages of beer production. Potential draft-line contamination is an under-tested but vitally important part of consumer exposure. The study combines DNA sequencing surveys with laboratory incubations well. The study utilizes deep sequencing appropriately and rightly avoids overly conclusive causations/associations/comparisons. This proper survey of the microbes present and their relative abundancies shows 'rampant' contamination, but high 'flux' and potential stochasticity in draft dispensed beers. Laboratory microcosms were set up to provide multiple niches for microbial propagation to survey microbes that may thrive in the draft system as planktonic or within a biofilm. This presented survey of information, including key prevalent microbes, may become important in future development of quality procedure development.

Review of text details:

Larger critical edits:

- The temperature utilized for incubation must be uniformly referred to throughout the manuscript. Whether it was either 20 C or 25 C or a range 20-25 C, it should be stated uniformly throughout.

Corrected.

- The sterile controls are mentioned in the methods (line ~446) but are not mentioned later in the manuscript. Their results are vitally important and are worth stating to validate the study. Were they analyzed, in what way, what was the result?

Those details are now presented as the closing sentence of the first result section.

- Details about beer 'habitat' and production methods are necessary to appreciate the study more and to have more potential follow-ups. This crucial information should be mentioned in the sample collection and culturing section of the methods. Examples of desired information (if known): Are beer(s) are filtered/centrifuged/pasteurized/barrel aged or not? What is the approximated IBU of the beers? What is the approximate ABV of the beers? Did any of them come from the same production facility? Is there anything known about the yeast(s) used in their productions? Any other researched details are worth noting.

This information is now included in the methods. We elected not to provide these data in the original version because we did not confirm the values and there is no substantiated source for IBUs (also, did not have access to brewery samples). Values for these are now presented in the Methods and were taken either from the supplier's web pages. Citing the sources will reveal the identities of the beers, which we need to avoid for this study.

- Details about beer dispensing 'environment' should be elaborated to appreciate the study more. This crucial information should be mentioned in the sample collection and culturing section of the methods. Examples of desired information (if known and willing to be divulged): What was the make-up of the lines: plastic type and diameter? Is this considered a long or short draft/approximate length of lines? What was the approximate temperature of the beer and/or keg storage? What was the standard cleaning procedure and schedule for such? What was the night storage within the lines? Were the lines allowed to flow or not prior to sample collection?

The methods were updated to include these details. We do not know the cleaning schedules of those specific lines before either sampling date - nor did the restaurant: before the Year 1 sampling, a distributor was responsible for line cleaning, and only cleaned the lines of what they connected during that visit. They used only caustic and water. In the intervening time, the distributor was changed to one that no longer provided cleaning, so the establishment hired a third party cleaning service who cleaned every two weeks, and only cleaned lines with kegs that were at least ~1/4 full because they used the same keg to flush the lines. Therefore, without direct knowledge of the cleaning of each of the four drafts in this study, we did not present those data. Working with regional brewers & retailers was a real eye-opener; none follow a published guideline and some even minimize line cleaning for fear of putting 'off tastes' in the beers.

- The additional experiment that would directly strengthen the manuscript significantly is a DNA analysis of the beer without exposure to the draft lines. If this is possible for any of the beers it should be done; if it is not logistically feasible it should be mentioned as such.

The Discussion was edited to include this consideration and the potential contamination sources.

Minor critical edits:

- Citations warranted:
 - o line 75 citation(s) warranted to support industry's revisitation

added

o line 80 citation(s) warranted to support the primary source of spoilage

added

o line 380 citation(s) warranted, even if unpublished, to support the statement that “we have not identified any...”

Changed wording to indicate how we tested and how many sites.

• Word choice/grammar/elaborative suggestions:

o Line 59 “go to waste” (word choice, overtly strong – ...are diminished? ...are significantly challenged?)

Changed wording to "are less effective".

o Line 76 “is to develop create products” (grammar – develop creative products? Develop and create products?)

corrected

o Line 101 “abundance of a particular microbe’s genome” (word choice, whole genome sequencing vs. amplicon sequencing – abundance of a sequence stretch of a particular microbe’s genome? Abundance of a particular microbe’s genomic region? Abundance of a particular microbe’s genomic taxonomic marker?)

Edited to remove reference to genomes because not all technology detects them.

o Line 121 “relative abundances of the microbial genomes” (word choice, whole genome sequencing vs. amplicon sequencing – relative abundances of the microbial taxonomic genes?)

Edited to be specific - ribosomal genes.

o Line 123 “nutrient-rich growth environment” (word choice, as finished beer isn’t generally considered nutrient-rich – potentially habitable growth environment? An appropriate proxy environment?)

Edited to emphasize fresh beer.

o Line 148, 448, etc “mixed” (elaborate how the cultures were mixed, every time it is mentioned – mixed via inversion? Mixed via vortexing?....?)

edited all instances.

o Line 320 “first draft pours of the day” this elaboration isn’t just a discussion point, this is a notable methodical point. It is worth mentioning elsewhere as well; for example, it is worth mentioning in methods line ~426.

This sentence and the methods were edited to provide a rationale for sampling the first draft.

o Line 325 “long culturing stage” (word choice, “long” is relative and two weeks is short to many investigations – two-week culturing stage?)

edited

o Line 340 “these bacteria are usually anaerobes” (word choice, as these are primarily facultative organisms and the environment impacts their current metabolism – commonly metabolize as anaerobes?)

edited

o Line 345 “In another study, Although these” (grammar and confusion)

corrected

o Line 354 “indicates that it is likely a” (word choice – it is maybe/possibly?)

edited

o Line 399 “A. cerevisiae strain we obtained” (elaborate with strain number, or some identifier)

Added the strain and source.

o Line 427 the beer type abbreviators, L, I, H should be reiterated

edited

Review of figures

• Figure representation:

o I think the figures being regrouped into DNA extraction type would help further understanding and flow. Figure 1, methods, should remain figure 1 (highly suggested elaborations/completions are below).

Figure 2 should be of starter sample DNA extractions, to establish a wholistic baseline; this would compile the current figure 2B and figure 4. Although probably motivated by logical domain categorization, as is, the current figure 4 (of starter DNA fractions) reads

as a somewhat disorienting afterthought. The current figure 2A should be its own figure, as figure 3, as it starts to compare starter to incubated DNA assemblages. The current figure 3 would then become figure 4 that compares biofilm and planktonic fractions from incubated DNA fractions.

This new figure order would flow nicely as the methodic figure 1 lays them out. The labeling of DNA fractions should be well established within figure 1 and verbiage should be reused exactly as figure headers/titles in the later figures. (e.g. starter DNA, incubated DNA).

Figure 1 revised, figures 2 & 4 combined, manuscript restructured as suggested.

- Figure 1 and legend:

- o The last sentence of legend, lines 544-546, applies to both A and B of the figure so should be present prior to the subpart descriptions, if kept as distinct subparts of the same figure.

Revised figure now has one panel.

- o A combined figure showing the connectivity of the 'field sampling' and the 'lab experiments' would accomplish the methodic story better than distinct subfigures.

Figure 1 edited.

- o If a methods figure is being utilized, more details should be included so that the details of the whole procedure can be understood at a glance and be referenced to when digesting the paper as a whole. This could aid the understanding of the future figures. A key missing aspect is the depiction of the 3 different sources of DNA material, which are later compared (starter DNA, incubated planktonic DNA, and incubated biofilm DNA).

- Extract DNA → extract starter DNA

- Sterile beer → sterile lager beer

- Depict some details from within your methods section alongside the arrows (e.g. 10x concentration via centrifugation)

- An arrow/arrows should be drawn from the bottom of the currently part A of figure (start cultures) to the initial culture

- The figure states 20 °C incubation, but the text of the manuscript states 25 °C – fix for uniformity, whichever is accurate

- Continue depicting the rest of the manuscript's experiments; it currently ends with incubation and doesn't explain the crucial DNA subfractions that comes from them.

- Draw two arrows from the mature culture tube and depict planktonic and biofilm fractions, and then depict DNA extraction of each

Each issue addressed.

- Figure 2 and legend

- o Lines 554 that defines the asterisk, should be elaborated; >1% of what?

“>1% of total reads” “>1% of total of starter DNA” “>1% of total cultured DNA” or similar verbiage

clarified

o 2A, bushes: the color variation of black, blue, red are indistinguishable on grey scale (and impaired vision); underlines or denotation of some kind could help accessibility

clarified

o The titles of figure 2B should state ‘starter’ along with the year sampled

edited

• Figure 3 and legend:

o 3A & 3C, bar graphs: the color variation of green, blue, red are indistinguishable on grey scale (and impaired vision); patterns or denotation of some kind could help accessibility

clarified

o Refer to incubated DNA appropriately

edited

• Figure 4 and legend:

o Line 574, “fungi present in the samples” – should be elaborated to “present in the starter samples”

edited

o The lines of text from ~267 should be reiterated in the figure legend to help the reader understand why there isn’t a chart for year 2

edited

o The sentence currently in figure 4 legend on line 578, “the dominant...” Is a discussion point and hypothesis and not exactly appropriate for a figure legend.

removed

o The titles of figure 4 should state ‘starter’ along with the year sampled

edited

Reviewer #2 (Comments for the Author):

Major issues:

In the abstract, the authors make the claim that the methods used here serve as "as starting point for efficient monitoring of beer spoilage..." (Line 33-34) and suggest that their data regarding biofilm growth will help define more appropriate draft system cleaning protocols.

There was no claim that the manuscript methods could be used to monitor spoilage. We claimed that the experimental data provides a starting point to understand which microbes are present and which can establish biofilms. We provided suggestions for line cleaning improvements in the Discussion. The abstract was edited to clarify these points.

There are a number of issues with these claims. First, draft beer spoilage is not necessarily defined by the presence of viable spoilage organisms. Instead, off flavor and aroma characteristics are the most important trait that would indicate altered cleaning regimes. The authors should indicate if the samples used had off flavor characteristics, either by sensory analysis (blinded triangle testing, as an example) or through more quantitative means (Such as HPLC, etc).

We have not found references to "beer spoilage" that do not involve rogue microbial activity (viability). In a retail setting, there is no unspoiled reference, which is a point of the manuscript that we hope more sellers become aware of. Consumers have no access to a 'correct' beer sample for comparisons, so they may dismiss a beer because it has a disagreeable aroma or flavor caused by spoilage microbes. These points were conveyed in the last paragraph of the Discussion.

Second, culturing samples is not overly useful to an establishment or brewery, as it provides a retrospective analysis of what was present at some point in the past-by the time this protocol has been followed, the contamination problem could have gotten worse to the point that more serious cleaning protocols are required.

We made no suggestion that an establishment or brewery should culture samples. We agree that if major contamination occurs after distribution, brewery-side monitoring is moot.

Last, typical draft cleaning protocols are expressly designed to disrupt biofilms and eliminate bacterial and fungal contaminants. For example, cycles of acidic and basic solutions are pumped through the lines using techniques to introduce turbulent flow intended to disrupt organic buildup/biofilm as well as any mineral scale that may be present.

This would be ideal, but the practice/protocol is not followed in many retail settings because of downtime, beer loss, and untrained staff. Outside of breweries, the Brewer's Association and their guidelines are generally unknown to bar staff. We hope this manuscript provides data to inform retailers that cleaning & maintenance regimens are important and may increase sales.

If the authors wish to make the claim in line 33-34, a more thorough discussion of current draft beer monitoring, cleaning, and sanitizing protocols must be included in this manuscript, along with a discussion of how their outgrowth protocol will specifically be useful to the various businesses within the draft beer supply chain.

We did not suggest the outgrowth experiments should be used for monitoring. We are currently using similar experiments to evaluate cleaning protocols and reagents that may improve retail procedures, but without those studies, no updated cleaning reagent suggestions can be made.

One could argue that this paper is more specifically assessing the line cleaning regimen of the particular institution from which the samples were taken, rather than a more general assessment of beer spoilage organisms. Is it known how (or if) this establishment was cleaning their draft lines, and if said protocols were in line with industry standards? How long had it been since the last cleaning? Is it known if line cleaning protocols changed between year 1 and year 2, or if the lines were cleaned the same time ago or each sampling?

This issue was addressed in response to reviewer 1. Retailers are under no obligation to follow any beer line cleaning standard, they are not even inspected for compliance by health inspectors. We added some description that contamination is a broader and common issue. In a separate location (not related to this study), we monitored (by plating) the abundance of bacteria after line cleaning: once by a large commercial vendor who only cleaned their own beer lines when they delivered; and once by general distributor (truck driver / keg delivery) when they cleaned a craft beer line at the insistence of the restaurant owner because the craft brewer himself demanded they pull his kegs from service because of on-site spoilage. The colony counts dropped substantially after cleaning; however, within two days, their numbers increased over a hundred fold. One could visibly observe biofilm being flushed out of the lines as a yogurt-like substance. This was the pilot study that inspired us to undertake this research.

Why was the lager beer chosen for the growth medium?

It was the only beer in the study for which a pasteurized, canned reference sample could be obtained. Also, we verified that it (the canned version) did not produce bacterial PCR amplicons and have included that result in the revised manuscript. In addition, it is a major brand and we can reliably obtain canned material for growth studies. We initially focused on its contamination because it was commonly spoiled on several retail taps and because it was the highest volume sale at two retailers, one of which was the site for this project.

How might the relative levels of inhibitory compounds (EtOH, hop compounds, etc) between the different beers alter the outgrowth of microbes from the different samples? Could the changes in outgrowth observed be explained by this? A discussion of the similarities and differences between lager-lager outgrowth and (for example) ipa-lager outgrowth should be included

We expected this would be the case; yet, the most significant changes (between biofilm and planktonic) were observed in the lager outgrowth experiment. We added the reported ABV and

IBU details of the unspoiled beers in the revised methods. We also added a new paragraph to the discussion regarding beer chemistry and bacterial tolerance.

Growth temperature vs draft line temperature: The authors chose an incubation temperature of 25 degrees C. Most draft systems operate at ~4 degrees C (or less) to ensure CO₂ does not break out of solution and to limit contaminant growth. The authors note that the diversity of the outgrown samples changed dramatically from the initial "starter". How much of this difference might simply be explained by growth rate differences at the increased temperature chosen for culturing?

We did not test the temperature dependence of the resulting community structures because of limitations of resources and the experimental setting. A focus of this work was the retailers, who may dismiss the importance of storage or draft line temperature (in this retailer's setting, the lines run at ambient from the keg room to the tap chillers). A temperature impact project is planned because we observed some plated bacteria grow well at 4 C.

Minor Issues:

The authors make the point that growth as a biofilm is important to the persistence of beer spoilage organisms in draft systems. However, the authors did not directly sample surface adhered/biofilm organisms from the draft lines, instead resorting to liquid samples. This likely will bias the initial abundance of organisms as well as the resulting outgrowth. A discussion of these biases should be included.

Now included as a discussion point.

The authors indicate that they sampled from the first pour of the day, after beer sat stagnant in the draft lines overnight. Typically, this first pour (Or more) is discarded until this lines have been purged to ensure that stagnant beer with greater potential for off flavor characteristics (due to spoilage, temperature differences, co₂ loss, etc). As such, the sample used for these experiments would not be representative of product served during service. A discussion of why this particular sample was used should be included.

We agree, and this would be great; however, this protocol is not commonly practiced because bar tenders are instructed to not waste beer. One of the authors is a brewer who has worked in two restaurants and two breweries with bars and was not instructed to flush lines at any location. A line can hold several pints and the mangers don't want a \$20-30 loss per tap per day. We edited the methods to indicate that we chose to sample first pours because that is what this retailer does. Again, we hope to change hearts and minds, but they need the data.

Discrepancy between text and figure: In figure 1B an incubation temperature of 20 degrees C is shown. In the text, line 448 an incubation temperature of 25 degrees C is shown.

This error was corrected.

November 3, 2021

Dr. Sean D. Moore
University of Central Florida
Burnett School of Biomedical Sciences
4110 Libra Drive
BMS 125
Orlando, FL 32816

Re: Spectrum01404-21R1 (Microbial communities in retail draft beers and the biofilms they produce)

Dear Dr. Sean D. Moore:

Your manuscript has been accepted, and I am forwarding it to the ASM Journals Department for publication. You will be notified when your proofs are ready to be viewed. Congrats to you and your team on a very cool paper!

Sincerely,

Jeffrey Gralnick
Editor, Microbiology Spectrum

Journals Department
Supplemental Dataset: Accept
Supplemental Dataset: Accept
Supplemental Dataset: Accept
Supplemental Dataset: Accept
Supplemental Dataset: Accept